# Numerical Analysis and Experimental Studies on the Residual Stress of W/2024Al Composites

**DOI:** 10.3390/ma12172746

**Published:** 2019-08-27

**Authors:** Guosong Zhang, Changhui Mao, Jian Wang, Ning Fan, Tiantian Guo

**Affiliations:** Institute of advanced electronic materials, General Research Institute for Nonferrous Metals, Beijing 100088, China

**Keywords:** W/2024Al composite, residual stress, heat transfer coefficient, constitutive equation, finite element method (FEM)

## Abstract

W/2024Al composites can be used for radiation shielding with desirable mechanical properties such as high strength, excellent corrosion resistance, and low density. The quench-induced residual stresses in W/2024Al composites were studied by experimental measurements and numerical analysis using ABAQUS software. Due to the accurate calculation of heat transfer coefficients and the established constitutive equation for description of the variation of yield stress at elevated temperature with different strain rates, the prediction of residual stresses in as-quenched composite blocks achieved by finite element method (FEM) is reliable. Moreover, X-ray diffraction and crack-compliance method were carried out to measure the stresses that developed at the surface and interior of the composites to validate the simulation results. Quenching residual stresses of composite blocks were investigated by taking the influence of quenching medium temperature into consideration. In addition, a comparative study on residual stress magnitudes of as-quenched 2024Al and W/2024Al composites was conducted, and the results show that stress magnitudes of W/2024Al composites are lower than that of 2024Al due to lower thermal gradients during the quenching process.

## 1. Introduction

The W/2024Al composite is a type of particle-reinforced aluminum matrix composite (PRAMC) used as a noble candidate for radiation shielding with desirable mechanical properties such as high strength, excellent corrosion resistance, and low density [1,2,3,4]. In the production process, the mechanical properties of PRAMC can be further improved by use of a high-temperature solution followed by rapid quenching and subsequent aging treatment [5,6]. However, an unfortunate consequence of heat treatment is the introduction of high-magnitude residual stresses because severe thermal gradients arise when quenching rapidly from the solution heat treatment temperature [7]. Generally, through water quenching, residual stress will arise within the W/2024Al composites due to two reasons. The first reason is attributed to the thermal expansion difference between the aluminum matrix and reinforced particles [8,9,10]. Secondly, the thermal gradients produced by non-uniform quenching cause inelastic strains that lead to residual stresses within the component [11,12,13]. Residual stress is known for inducing premature failure through cracking, reducing fatigue strength, evoking stress corrosion or hydrogen cracking, changing the mechanical properties of materials and causing distortion and dimensional variation [14,15,16,17]. 

Owing to the importance of thermal residual stress itself in aluminum alloys, many studies on quench-induced residual stress have been undertaken in the last few decades. Numerical analysis, like employing finite element method (FEM), and experimental methods such as X-ray [18], crack-compliance method [19], and hole drilling [20,21,22] are commonly used for the research of residual stresses developed during the quenching operation for aluminum alloys. Specifically, X-ray diffraction is one of the widely used methods to determine residual stresses due to its reliability and convenience, though the method has the drawback of being unable to determine the through-the-thickness distribution of residual stresses. On the contrary, the crack-compliance method (CCM) generally provides excellent spatial resolution of residual stress profiles [23]. 

The finite element method (FEM) technique is proven to be a feasible and cost-effective alternative because of its versatility, accuracy, and efficiency. Caner S [24] developed an FEM model based on a mathematical model capable of predicting temperature change, evolution of phases, and internal stresses for complex 3D engineering components. Li [25] proposed the FEM model which was coupled with temperature, phase transformation, and stress/strain, and the computer simulations of the heat treatment process of work pieces corresponded with the experimental and analytical results. The model established by Xiao [26] to simulate the quenching process of aluminum alloys can be adopted to predict distortion and residual stress of complex shape aluminum castings during quenching with high accuracy. In order to make representative predictions of transient temperature distribution and the resultant stress distribution upon quenching of sample, the time-dependent heat transfer coefficients (HTCs) between quenched component and quenchant are required. The report [27] indicated that a 10% deviation in HTCs can lead to a 30% deviation in stress magnitudes in the numerical analysis. The inverse heat transfer method was frequently used in the calculation of the HTCs [28].

The magnitudes and distributions of residual stresses in different kinds of aluminum alloys have received intensive attention in the last decades, while quantitative investigations of the residual stresses in PRAMCs remain limited. Therefore, the quantitative characterization of residual stress for W/2024Al composites by using numerical simulation and experimental method was the impetus for this study. 

## 2. Materials and Methods

### 2.1. Materials and Heat Treatment

2024Al–6vol.%W(p) composites (index p denotes particles) were prepared by powder metallurgy method through power mixing, cold isostatic pressing (CIP), vacuum degassing (VD), hot isostatic pressing (HIP), and hot extrusion. The experimental geometries considered consisted of several 100 × 78 × 36 mm^3^ blocks were cut from W/2024Al extruded profiles. The heat treatment process implemented in this paper is described in Figure 1. The composite blocks considered here were firstly heated in a furnace to 470 ± 5 °C for 45 min, then promptly transferred into agitated 10 °C water.

### 2.2. The Determination of Heat Transfer Coefficients

Jominy end-quench tests were carried out to obtain heat transfer coefficients (HTCs), and the schematic diagram of the experimental facility (General Research Institute for Nonferrous Metals, Beijing, China) is shown in Figure 2. The ϕ48 × 150 mm cylinder specimen was isothermally treated at 470 °C (the temperature of solution treatment) until the whole specimen reached the heat equilibrium, then the cylinder specimen was rapidly transferred to the fixer of test device where it is held vertically and sprayed with a controlled 13.9 L·min^−1^ flow of water onto the end of the sample. To monitor and record the transient temperature response of point A and B, as shown in Figure 2 on the sample’s central axis, two thermocouples were embedded perpendicularly into the sample at point A and B, which are respectively located at 20 and 50 mm away from the quenched end. 

Then, the initial HTCs can be calculated by using the inverse heat method and Newton’s law of cooling based on the ‘temperature–time curves’ of point A and B. The temperature of point O can only be calculated and not directly measured, and the calculation process is shown as follows. 

The temperature of point O during the end-quench test can be described as
(1)$$T\left(O,t\right)=2T\left(A,t\right)-T\left(B,t\right)+\frac{{\left(\Delta x\right)}^{2}c_{1}\left(T\right)\rho_{1}\left(T\right)}{\lambda_{1}\left(T\right)}\frac{T\left(A,t+\Delta t\right)-T\left(A,t\right)}{\Delta t}\;-\frac{d\lambda \left(T\right)}{dT}\frac{T^{2}\left(B,t\right)-2T\left(A,t\right)T\left(B,t\right)+T^{2}\left(A,t\right)}{\lambda_{2}\left(T\right)},$$
where T(x,t) is the temperature of points in the central axis, T(O,t), T(A,t), and T(B,t) represent the temperature of the point O, A, and B during the end-quench test. 

The specific heat capacity c(x,t), density ρ(x,t), and thermal conductivity λ(x,t) are all a function of T(x,t), where $c_{1}\left(T\right),\left(T\right),\;\lambda_{1}\left(T\right)\;$ represent the average of values at T(A,t+Δt) and T(A,t), respectively. While $\lambda_{2}\left(T\right)$ is the average value of thermal conductivity at T(A,t) and T(B,t), Δx is the distance between the point A and B.

The equation for heat flux of point O during end-quench test can be written as
(2)$$q_{T\left(O,t\right)}=\lambda_{3}\left(T\right)\frac{T\left(A,t\right)-T\left(O,t\right)}{\Delta x_{1}},$$
where $\lambda_{3}\left(T\right)$ is the average of thermal conductivity at T(A,t) and T(O,t), and $\Delta x_{1}$ is the distance between point A and O.

Then, the values of HTCs can be obtained by Equation (3).
(3)$$h_{T\left(O,t\right)}=\frac{q_{T\left(O,t\right)}}{T\left(O,t\right)-T_{Q}},$$
where $T_{Q}\;$ is the temperature of the quenchant.

In this paper, the HTCs $h_{T\left(O,t\right)}$ calculated by Equation (3) are “initial HTCs”. The values are not the ultimate results due to errors that inevitably occur in the experimental and calculative processes. Thus, initial HTCs need to be amended by an inverse heat transfer model in DEFORM software where initial HTCs are imported as primary trial values through iterative calculation. The authors do not give further description for iterative calculations here. We adopted the approach combining the end-quench test and DEFOR software, not only for time savings but also accuracy improving.

### 2.3. Hot Pressure Test for Characterization of Mechanical Properties

To present the mechanical properties at different temperatures, cylindrical W/2024Al samples with φ8 × 12 mm were conducted on the Gleeble thermomechanical test machine (GLEEBLE 1500, DATA SCIENCES INTERNATIONAL, INC, St. Paul, MN, USA) at temperatures ranging from 50 to 450 °C with strain rates of 0.001, 0.01, 0.1 and 1.0 s^−1^. Then, the compressive deformation behaviors of W/2024Al composites were investigated by creating constitutive equation based on the hot pressure test data. 

### 2.4. FEM Modeling of Quenching Process and Residual Stress Measurement

In this paper, the commercial software package ABAQUS (ABAQUS 6.14, Dassault Systèmes, Paris, France) was used to develop a simulation model with W/2024Al composites block during quenching. As shown in Figure 3a, the FEM model of this problem only includes one-eighth of composite block to reduce CPU time for simulation, due to its geometry and boundary conditions. The studied paths were established as shown in Figure 3b to investigate the stress distribution of as-quenched block, the stress components S11, S22, and S33 are parallel to the L (longitudinal), LT (long transversal), and ST (short transversal) direction, respectively. The stresses are assumed to be uniformly zero throughout the block sample before the quenching process in this model. Quenching is a multiscale process which couples physical phenomena that occur simultaneously, such as heat transfer, phase transformation, and plastic deformation [29]. W/2024Al composites should be quenched from the solution-treatment temperature as rapidly as possible to minimize the precipitation phase in the composite. Due to the metallurgical properties of aluminum matrix composite, it is generally assumed that no phase change takes place during the quenching process. Thermal stress is mainly generated by the distribution of different temperatures in different parts of the work piece during the quench operation. Due to the authors’ decision to ignore the phase transition to simplify the simulation model of quenching, which is reasonable, a sequential analysis of coupled temperature–stress was conducted in this paper. Nodal temperatures were calculated in a heat transfer step by importing the HTCs as prescribed boundary conditions. The stress analysis was then obtained in a static step by importing the temperature history as a prescribed predefined field with appropriate boundary conditions specified on the symmetry planes. The C3D8R and D3CD8 element in the ABAQUS library were selected in analysis of temperature and stress distribution simulation, respectively. The meshed element size was approximately 1 × 2.14 × 3mm^3^, which was considered fine enough according to the publication [30]. 

In order to validate the simulated results, residual stresses that developed in the surface and interior were measured by X-ray method and crack-compliance method, respectively. For the crack-compliance method, a planar slit is introduced by incrementally cutting into a rectangular block containing quenching residual stresses in steps of increasing depth [31]. The calculation of the residual stress field existing normal to the cut plane (the stress for S11 component on Z path) prior to slitting can be calculated by combing finite element method based on the strain data collected from strain gauge, as shown in Figure 3a. The detailed calculation description of this method was not given here because the method was frequently reported in many reports [32].

### 2.5. Research Procedure

As previously mentioned, this paper involves the determination of heat transfer coefficients and establishment of constitutive equations in order to set up an FEM model with high accuracy for stress simulation. In addition, the influence of water temperature on the residual stress distribution of blocks after quenching was also investigated in this paper. Moreover, the comparative study on stress magnitudes and distributions of as-quenched 2024 alloys and W/2024Al composites was conducted. In order to clarify the research process of this paper, a flowchart of the research procedure is given in Figure 4.

## 3. Results and Discussion

### 3.1. The Results of the Heat Transfer Coefficients 

Firstly, the temperature T(O,t) and heat flux $q_{T\left(O,t\right)}$ of point O in the quenched end can be obtained by Equations (1) and (2). The cooling curve and heat flux of the surface quenched by 10 °C water are shown in Figure 5a, and it can be seen that the surface temperature first drops rapidly, then decreases gently until becoming constant, while the heat flux increases promptly to the peak at the beginning and subsequently decreases sharply followed by a slower downward trend. 

Secondly, DEFORM software simulation was carried out to amend the HTCs through iterative calculation. Figure 5b shows experimental and simulative temperature of point A and B in cylinder sample used in end-quench test when 10 °C water was adopted as quenchant. The temperature results of two points from the FEM simulation are very consistent with those from experimental data. Therefore, the temperature distributions simulated by the FE model agree well with the experimental results, which confirms that the HTCs obtained by the present work can be adopted as accurate parameters to give a reasonably precise estimate for the temperature field of W/2024Al composites during quenching.

For further research of the influence of water temperature on residual stress of blocks, a series of experiments and calculations were conducted to calculate the HTCs of water with different temperatures, as illustrated in Figure 6, which indicates that the HTCs are functions of the water temperature and HTCs curves of different temperature share the similar trend as surface temperature of block changes during quenching.

### 3.2. Constitutive Equations for Representing Mechanical Properties of W/2024Al Composites

The material constitutive behavior was characterized by creating a constitutive equation which is commonly incorporated into the FE model for deformation simulation. According to the true stress–strain curves at various temperatures with different strain rates, the flow stress changes indistinctly with the strain rate when the temperature is below 300 °C. Therefore, the constitutive equation was established derived on the true stress–strain curves ranging from 300 to 450 °C at different strain rates. After a series of calculation, the hyperbolic sine equation can be written as
(4)$$\varepsilon =3.03\times 10^{9}\cdot \left[\sinh(0.011844\sigma \right){]}^{4.96847}\exp(\frac{-139823}{RT}).$$

The constative equations for the materials are conducted through interface for user subroutine UHARD provided by ABAQUS software. In order to verify the developed constitutive equation for W/2024Al composites at elevated temperatures with strain rates from 0.001 to 1 s^−1^, a comparison between the experimental and calculative results is illustrated in Figure 7, the dash and solid lines represent the calculative and experimental values of flow stresses, respectively. The calculated values are slightly higher than the values obtained from experimental results when the strain rate is 0.01 s^−1^ and greater differences appear in calculated and experimental values at strain rate of 1 s^−1^. However, on the whole, the results indicate that the proposed deformation constitutive description for yield stresses gives a reasonable estimate of the flow stress for W/2024Al composites, and can be adopted to analyze the problems during the quenching process. 

### 3.3. The Comparison between Simulated Results and Experimental Data

In this section, the X-ray diffraction method and crack-compliance method were utilized to determine residual stresses after quenching for the validation of simulated results. As illustrated in Figure 8, comparisons of surface stresses for the S11, S22 components between simulation results and experimental data on the Path X, Y are given. The simulated results indicate a condition of plane stress is assumed to exist in the surface of block, that is, no stress is assumed perpendicular to the surface. As the figures illustrate, the simulated values are consistent with the experimental results on the paths where an average deviation of 25 MPa was found, except for the region near the edge. This can be explained by the stress fluctuation area that exists near the edge due to the edge effect. Compared with the internal regions, the edges yield earlier because of the rapid cooling rate and larger quenching intensity. Therefore, the thermal stresses for one component in the edges decrease significantly and maintain a fairly low level since the edges have cooled down. Thus, high-level stresses for another component develop in the edges and high-level stresses generate in the region close to edges in order to balance the edge stresses [33]. The measured strain values and the calculated results of residual stress obtained by the crack-compliance method are expressed in Figure 9, and the residual stresses along the thickness reveal an “M” profile which was typical for the quenched blocks. At the beginning of quenching, the cooling down of the external regions with contraction and hardening occurs prior to internal regions, which induces the internal compressive stresses and is balanced by external tensile stresses. After a period of time, the internal materials start to cool down and shrink, whereas the shrinking process is restricted by the hardened outer layer, which gives rise to the internal tensile stresses and is finally balanced by external compressive stresses. The simulated values are consistent with the experimental measurements on the Z path, and it can be seen that the crack-compliance method and FEM can be applied for characterization of residual stress for W/2024Al composites in this investigation.

### 3.4. Influence of Water Temperature on Quenching Residual Stress of W/2024Al Composite Blocks

Quenching conditions, including the quenching medium and temperature prior to the quenching process, should be designed properly in the course of manufacturing [34]. The magnitudes of residual stresses can be restricted without any process of stress reduction if quenching conditions are appropriate [35]. In this part, the authors performed three simulations using ABAQUS software to investigate the effects of water temperature (20, 40, and 60 °C) on residual stress. Models with a dimension of 100 × 78 × 36 mm^3^ were selected. The HTCs of the different temperatures of water are shown in Figure 6 in Section 3.1. It was reasonable to select the HTCs of the 10 °C water to calculate the temperature field for water quenching with 20 °C because of the small temperature difference, although a certain degree of error could have resulted. The maximum stresses for the S11, S22, and S33 components at different water temperatures are given in Figure 10, and it can be seen that the maximum compressive stresses are higher than the maximum tensile stress for one particular stress component at different water temperatures. The reduction percentages of the maximum stresses were put forward to quantify the effects of water temperature on residual stresses. As the water temperature increased from 20 to 60 °C, the reduction of maximum tensile stresses for the S11, S22, and S33 stress components were 65.4%, 72.4%, and 43.5%, respectively, while the reductions of the maximum compressive stresses for the S11, S22, and S33 stress components were 30.7%, 28.6%, and 63.3%, respectively. The average reduction percentage of the maximum tensile stresses was greater than that of the maximum compressive stresses. 

The cooling rates of the points of maximum stresses in W/2024Al composite blocks quenched by water of different temperatures are illustrated in Figure 11a,b. It can be seen that the cooling rate of the point with maximum tensile stress decreases more than that of the point with maximum compressive stress as the water temperature was increased from 20 to 60 °C. Furthermore, the reduction of the cooling rate in the interior regions, where tensile residual stresses are generated, was much more obvious than that in the external regions where compressive stresses developed as the water temperature increased. Therefore, the relief effect of the tensile stresses achieved by increasing water temperature was more apparent. 

The predicted residual stresses contours for the S11, S22, and S33 components of the blocks quenched by water of different temperatures are shown in Figure 12, Figure 13 and Figure 14. It can be seen that the stresses contours for one particular stress component have analogous stress distributions, though the magnitudes of stresses decrease distinctly with increasing water temperature. The tensile properties of W/2024Al composites quenched by water of different temperatures are given in Table 1, with two samples were tested for each condition, and their average property value was adopted. As shown in the Table 1, the as-quenched composites possess favorable mechanical properties when treated with water of elevated temperature. The W/2024Al composites are tolerant of higher water temperatures due to having a low sensitivity to the quench rate. Thus, the alternative to reduce the quench-induced residual stresses by elevating water temperature during quenching is satisfactory and useful.

### 3.5. A Comparison of Residual Stress Magnitudes and Distributions among 2024 Aluminum Alloys and W/2024Al Composites after Quenching

As mentioned, 2024 aluminum alloys were selected as a matrix for the W/2024Al composites. A comparison of residual stresses of the two materials after identical solution and quenching treatments is worth studying for a better understanding of quench-induced residual stress. 

In this part, the crack-compliance method and ABAQUS simulations were used to assess the stress magnitudes of the two materials. For FEM, all the temperature and stress field simulation data of the 2024 aluminum alloys during quenching have already been published [36]. For effective comparison, the 2024Al and W/2024Al composite models shared the same dimensions of 100 × 78 × 9/18/36 mm^3^ (with thickness varying), and water at 60 °C was chosen as the quenching medium. 

The stress magnitudes measured using the crack-compliance method along the thickness in 2024Al and W/2024Al blocks (100 × 78 × 36 mm^3^) quenched by water at 60 °C are given in Figure 15, and it can be seen that the stress magnitudes of 2024Al are greater than those of W/2024Al composites. For the simulation results, the maximum stresses of as-quenched 2024Al and W/2024Al composite blocks quenched by water at 60 °C and with varying thicknesses are shown in Figure 16, Figure 17 and Figure 18, where all the maximum stresses of 2024Al are greater than those of W/2024Al. Moreover, as thickness increased, the differences in the maximum compressive stresses of the two materials tend to be less obvious, while apparent differences do appear in their maximum tensile stresses. Both the experimental and simulation results indicate that the as-quenched 2024Al alloys exhibit higher stresses than the W/2024Al composites. 

The yield stresses of as-quenched 2024 alloys and W/2024Al composites at different temperatures are given in Table 2. It can be concluded that the yield stresses of W/2024Al composites are greater than those of 2024Al alloys at various temperatures due to the particle-strengthening mechanism of the metal–matrix composites. Generally, the substantially higher yield stress of as-quenched W/2024Al composites makes them inherently more prone to residual stresses and machining distortion in relatively thick sections compared to the 2024 aluminum alloys. 

The different thermal gradients during quenching of 2024Al and W/2024Al composites cause the differences in stress magnitudes. Figure 16, Figure 17 and Figure 18 indicate that the peak residual stress of both materials at varying thickness is lower than the yield stress. In this case, stress development was controlled mainly by thermal gradients rather than as-quenched yield strengths in thick samples. Figure 19 gives the cooling curves of two specific points (A and B denote the points with the lowest and highest temperatures during quenching, respectively) in as-quenched 2024Al and W/2024Al blocks (100 × 78 × 36 mm^3^), it can be seen that the 2024Al experienced greater quenching severity than W/2024Al. Samples of 2024Al and W/2024Al composites have distinct thermal properties and, therefore, differential thermal gradients develop during quenching process. Moreover, the thermal gradients in the relatively interior regions of the two blocks with identical dimensions tend to vary as the thickness increases. The apparent differences in the maximum tensile stress of the two materials, as illustrated in Figure 16, Figure 17 and Figure 18. Based on the calculations, compared with 2024Al, the W/2024Al exhibits low stress magnitudes due to its lower quenching severity.

## 4. Conclusions

FEM analysis provided by ABAQUS software has been selected to investigate the stress distribution of W/2024Al composites under different quenching conditions. Owing to the accurate calculation of heat transfer coefficients and the established constitutive equation for description of the variation of yield stress at various elevated temperatures with different strain rates, the prediction of the residual stress of as-quenched W/2024Al blocks obtained by FEM is reliable. In addition, the quenching model was verified by experimental measurements. The influence of water temperature on stress distributions was investigated. A comparative study of residual stress magnitudes and distribution between as-quenched 2024Al and W/2024Al composites was conducted. The main results of this paper are as follows:(1)The reduction of cooling rate in interior regions where tensile residual stresses generated is distinct to that in the external regions where compressive stresses developed as quenchant temperature increases. Thus, the relief effect of tensile stresses achieved by increasing water temperature is more distinct. In addition, W/2024Al composites exhibit lower stresses with desirable mechanical properties when an elevated water temperature was adopted.(2)The comparative study shows stress magnitudes of as-quenched 2024Al are greater than those of the W/2024Al composites. The different thermal gradients mainly result in the occurrence of distinct differences in stress magnitudes of the two materials. W/2024Al exhibited low stress magnitudes compared with 2024Al due to lower quenching severity during the quenching process.

## Figures and Tables

**Figure 1 materials-12-02746-f001:**
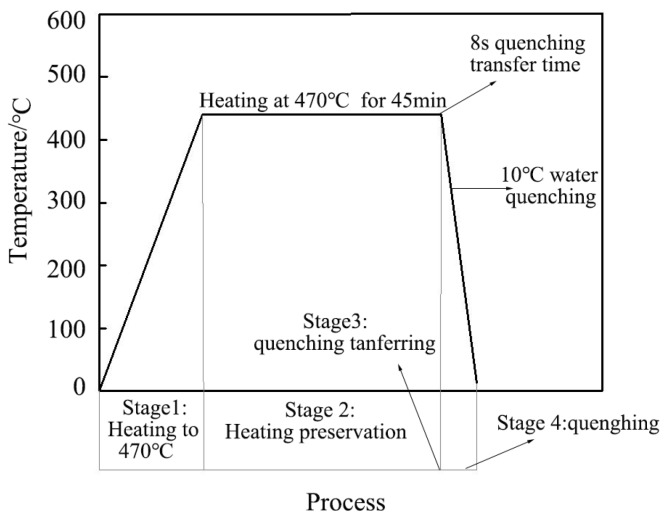
Schematic curve of solution heat treatment and water quenching process.

**Figure 2 materials-12-02746-f002:**
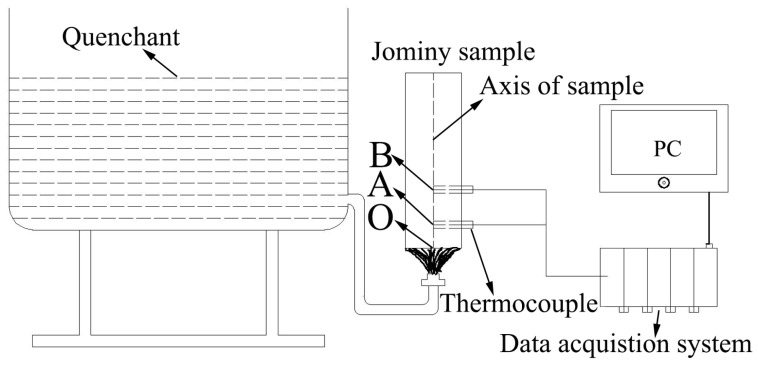
Schematic diagram of Jominy end-quench test.

**Figure 3 materials-12-02746-f003:**
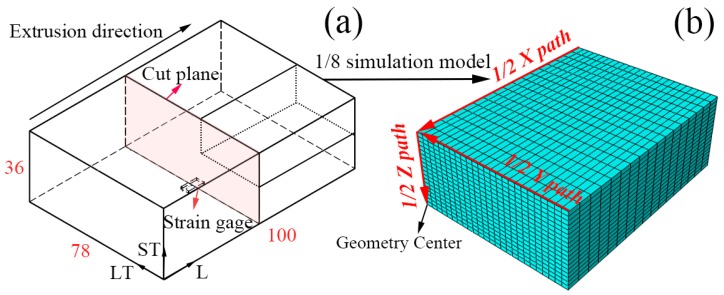
(**a**) Schematic view of crack-compliance method; (**b**) Mesh result and investigated paths.

**Figure 4 materials-12-02746-f004:**
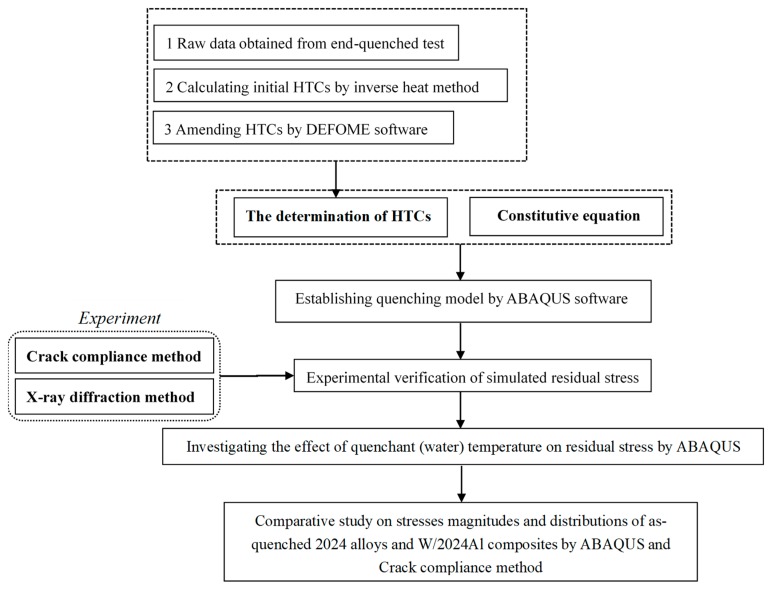
Flowchart of the research procedure.

**Figure 5 materials-12-02746-f005:**
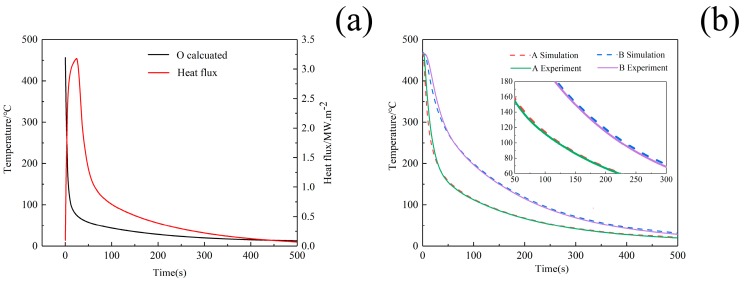
(**a**) Cooling curves and heat flux of surface in cylinder sample; (**b**) Comparison between simulated and experimental cooling curves of the point A and B during end-quench test when 10 °C water is adopted as quenchant.

**Figure 6 materials-12-02746-f006:**
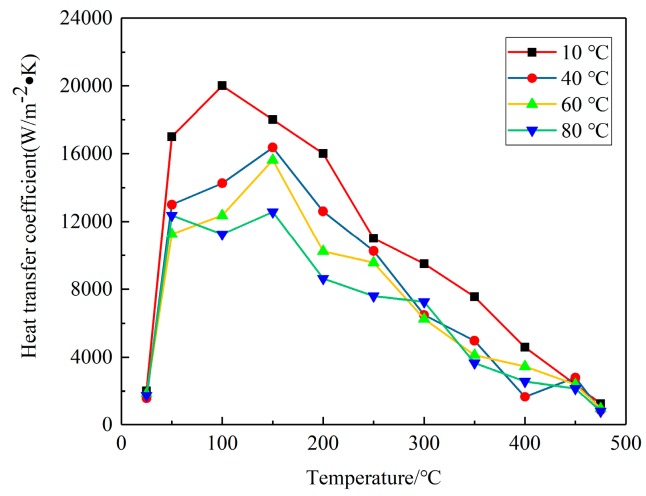
Modified heat transfer coefficients (HTCs) for water with different temperature.

**Figure 7 materials-12-02746-f007:**
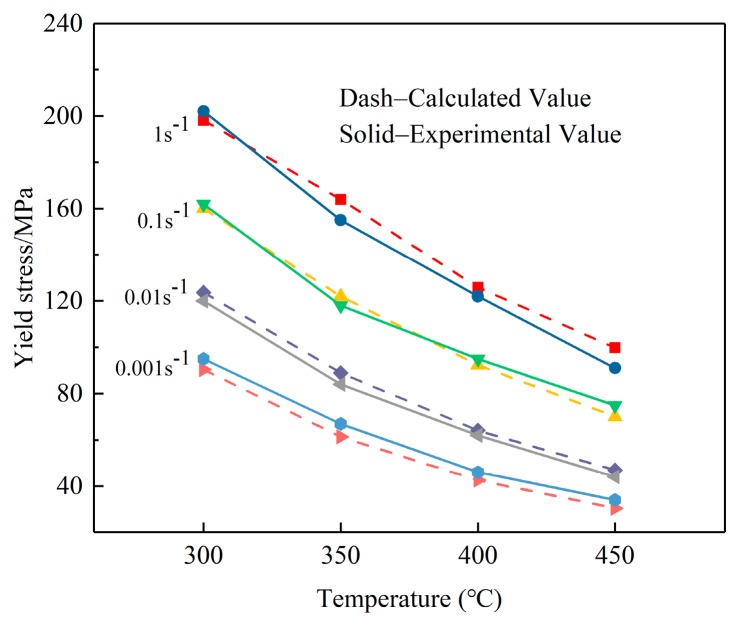
Comparison between experimental and calculated flow stress values of W/2024Al composites at high temperatures with different strain rates.

**Figure 8 materials-12-02746-f008:**
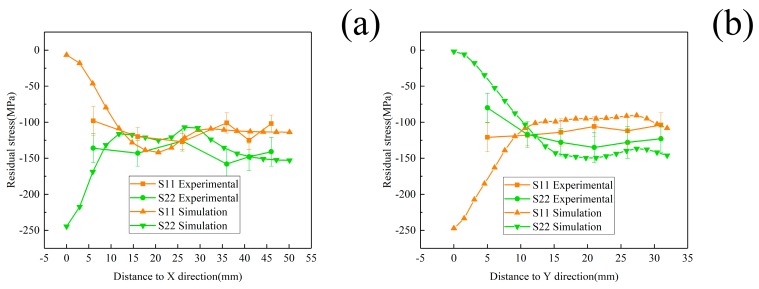
Comparisons of surface stresses distributions for the S11, S22 stress components between experimental data through X-ray diffraction method and simulation results on the paths. (**a**) 1/2 X Path; (**b**) 1/2 Y Path.

**Figure 9 materials-12-02746-f009:**
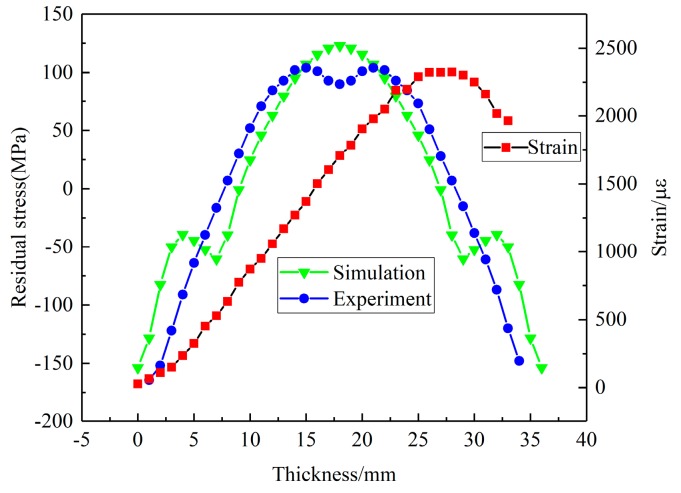
Comparisons of S11 component stress along thickness (Z path) between experimental data obtained by crack-compliance method and simulation results.

**Figure 10 materials-12-02746-f010:**
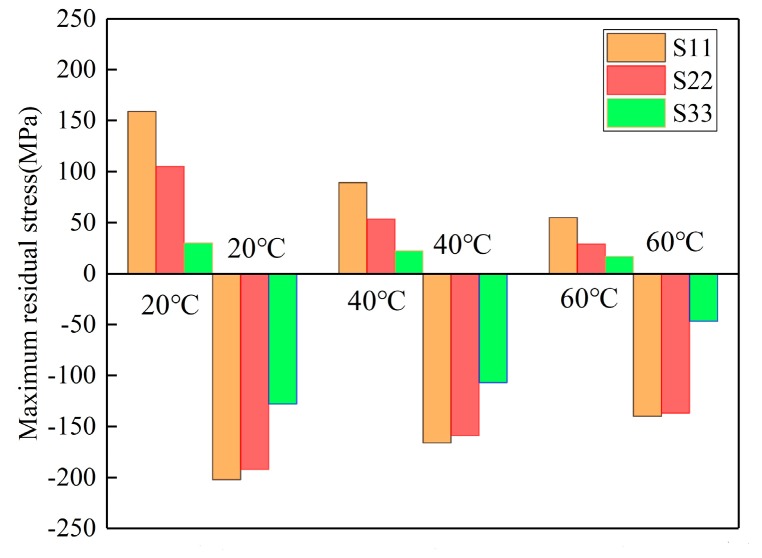
Maximum residual stresses of W/2024Al composite blocks quenched by water of different temperatures.

**Figure 11 materials-12-02746-f011:**
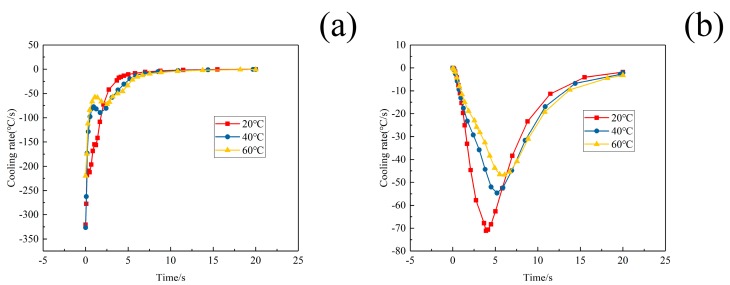
The cooling rates of the points of W/2024Al composite blocks quenched by water of different temperatures; (**a**) The point of maximum compressive stress; (**b**) The point of maximum tensile stress.

**Figure 12 materials-12-02746-f012:**
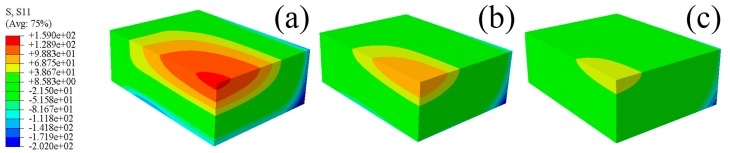
Predicted residual stresses contours of the 1/8 model for the S11 component; (**a**) 20 °C; (**b**) 40 °C; (**c**) 60 °C.

**Figure 13 materials-12-02746-f013:**
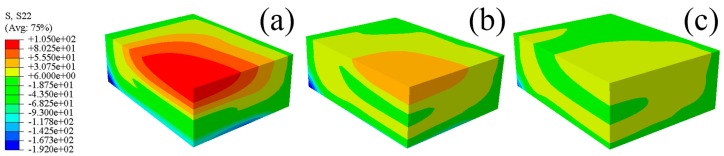
Predicted residual stresses contours of the 1/8 model for the S22 component; (**a**) 20 °C; (**b**) 40 °C; (**c**) 60 °C.

**Figure 14 materials-12-02746-f014:**
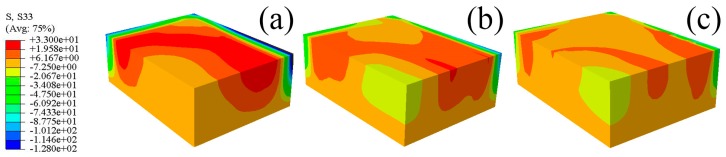
Predicted residual stresses contours of the 1/8 model for the S33 component; (**a**) 20 °C; (**b**) 40 °C; (**c**) 60 °C.

**Figure 15 materials-12-02746-f015:**
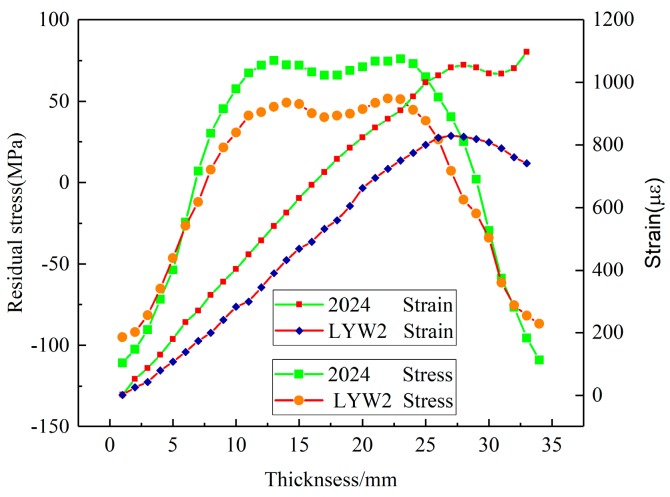
Residual stresses of 2024Al and W/2024Al samples quenched in 60 °C water measured using the crack-compliance method.

**Figure 16 materials-12-02746-f016:**
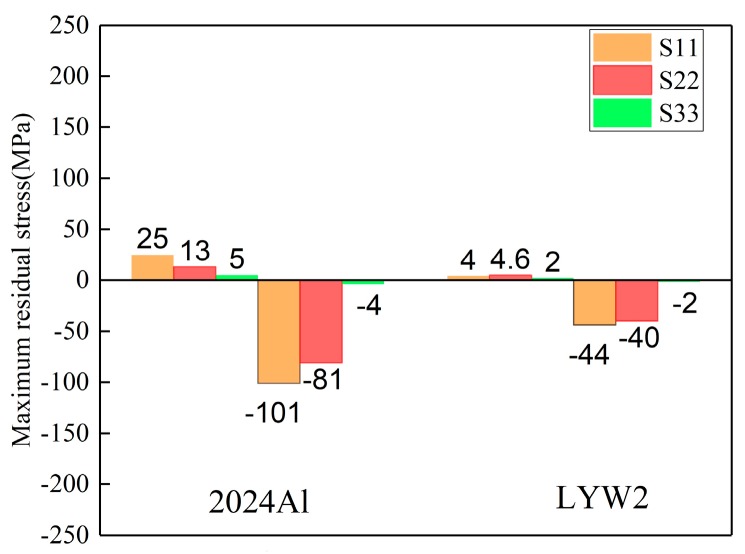
Maximum stresses of as-quenched 2024Al and W/2024Al blocks with the thickness of 9 mm.

**Figure 17 materials-12-02746-f017:**
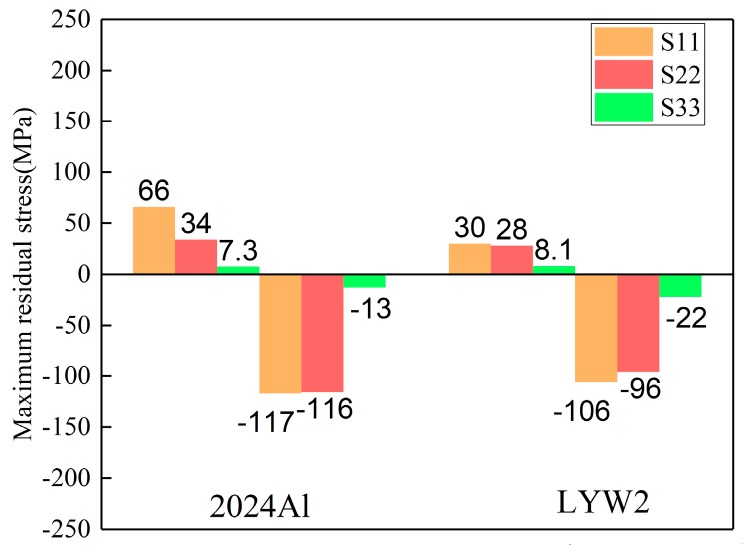
Maximum stresses of as-quenched 2024Al and W/2024Al blocks with thickness of 18 mm.

**Figure 18 materials-12-02746-f018:**
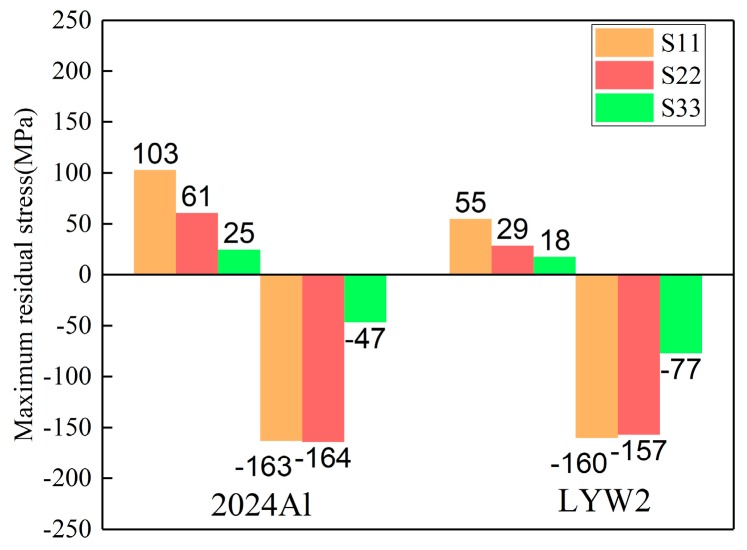
Maximum stresses of as-quenched 2024Al and W/2024Al blocks with the thickness of 36 mm.

**Figure 19 materials-12-02746-f019:**
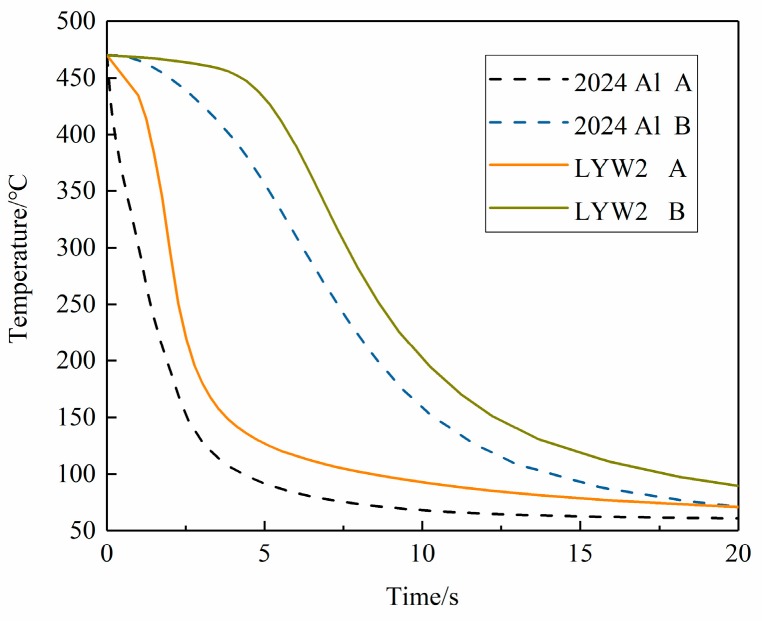
Cooling curves of 2024Al and W/2024Al block; A denotes the point with the lowest temperature; B denotes the point with the highest temperature during quenching process.

**Table 1 materials-12-02746-t001:** The tensile properties (MPa) of W/2024Al composite blocks quenched by water of different temperatures.

Water Temperature (°C)	Ultimate Tensile Stress (MPa)	Yield Stress (MPa)	Elongation (%)
20	350	218	11
40	355	223	13
60	343	210	10

**Table 2 materials-12-02746-t002:** The yield stresses (MPa) of as-quenched 2024 alloys and W/2024Al composites at different temperatures.

T (°C)	50	100	200	250	300	350	400	450	500
2024 alloys	182	155	116	99	83	72	41	34	25
W/2024Al	267	190	160	/	135	111	70	50	/

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
