# Peer review of "Numerical Analysis and Experimental Studies on the Residual Stress of W/2024Al Composites"

_materials, 2019, doi:10.3390/ma12172746_

Round 1
Reviewer 1 Report
Residual Stress of W/2024 Al Composites has been studied in this research. Although the topic is interesting and the research seems to conduct correctly, the manuscript should be revised to be acceptable for publication in Materials. Below is the most important issues that should be addressed in the revision:
1- The format of the manuscript should be changed entirely. In the current format, it is hard for the reader to understand the exact methodology of the modelling, simulations, and experiments. In some paragraphs of the results and discussions, the experimental procedure has been reported. Also, it is not clear which part of the research has been performed by ABAQUS, Deform, or experiments. Maybe, it is better to suggest a flowchart for the research procedure.
2- The English of the manuscript should be revised.
3- In line 121, what do you mean by “uneven temperature distribution”?
4- The reason for the ignorance of phase transformation is not clear.
5- Why the authors, used L, LT, and ST for the direction? What are the abbreviations stand for?
6- Please change the legend of Fig. 5. Numbers without any descriptions and any unit do not have any meaning.
7- Eq. (2) should be corrected? It cannot be read now.
8- Do you use both Deform and ABAQUS software? If so, please describe the simulation parameters for the Deform software. Also, please clarify that, which part of the simulations has been done by this software? Furthermore, please describe the reasons for using two different software.
9- For the confirmation of the simulation results, the authors used the previously published simulation results. If others have done the simulations successfully, then what are the benefit and the novelty of this research?
10- There is not any physical discussion about the results. The manuscript is more similar to a report, rather than a scientific paper. Please add a couple of sentences to discuss the logical relations between the results in terms of physical basis.
Reviewer 2 Report
The style of writing should be improved. Authors should not use the words "It’s obvious", but why present reasons, why they think so. Also, if something is obvious, why did they do it. The authors should indicate the novelty obtained as part of the work.
In introduction should be muched around the policy's problem that the authors want to resolve.
The descriptions of the axes in the graphs need to be refined, for example in Figure 1, instead of "time" there should be a "process". In addition, the presentation of units should be standardized.
Figure 5 shows the relationship for 10,40,60.80, what are the values. The text does not contain a description of the course of the results obtained.
In line 179, the authors give their own thermo-mechanical test machine name, which, according to the reviewer, is not appropriate. The machine should, however, be characterized by the accuracy of the parameters obtained in the test.
Line 187 In pattern 2, the letters overlap, the pattern should be corrected. Chart 6 should be commented on in the text. Especially that the curves are arranged in the order of 0.001; 0.1; 0.01 and 1s, not by value. For the 0.001 and 0.01 s lines, the calculated values are higher than the values obtained from experimental studies, which is on the safe side. However, for 0.1 and 1 s values are calculated higher and the difference between individual results increases for higher strain rates (for 10 s probably the difference would be significant). This observation should be commented on in the text. The authors should, therefore, consider giving the scope of the constitutive equation.
Line 237 (fig. 8) According to the reviewer, the results are not consistent, as the authors wrote in the text. The results are beyond the limits of laboratory test errors.
There is no information in the text whether they are individual results of the research or the average of several results.
Why the authors did not set the trend line and its equation. Comparison of the trend line with the results of the simulation would allow to quantify the differences for each point.
Line 302 (table 1) The numbers should be given in the same accuracy.
Line 342 Figure 16 is hard to see.
